# Sex disparities in the associations of overall versus abdominal obesity with the 10-year cardiovascular disease risk: Evidence from the Indonesian National Health Survey

**Fathimah S. Sigit**[1,2]*, **Dicky L. Tahapary**[2,3], **Woro Riyadina**[4], **Kusharisupeni Djokosujono**[1]

1 Department of Public Health Nutrition, Faculty of Public Health—Universitas Indonesia, Depok, Indonesia, 2 Metabolic Disorder, Cardiovascular, and Aging Cluster, Indonesia Medical Education and Research Institute, Faculty of Medicine—Universitas Indonesia, Jakarta, Indonesia, 3 Department of Internal Medicine, Dr. Cipto Mangunkusumo National Referral Hospital, Faculty of Medicine—Universitas Indonesia, Jakarta, Indonesia, 4 National Research and Innovation Agency, Jakarta, Indonesia

* fathimah10@ui.ac.id

**Data Availability Statement:** Data cannot be shared publicly because the authors signed a data usage statement with the Ministry of Health,

## Abstract

### Background

Cardiovascular diseases (CVDs) are a leading cause of disability-adjusted life years in Indonesia. Although obesity is a known risk factor for CVDs, the relative contributions of overall versus abdominal obesity are less clear. We aimed to estimate the 10-year CVD risks of the Indonesian population and investigate the separate and joint associations of overall and abdominal obesity with these risks.

### Methods

Using nationally representative data from the Indonesian Health Survey (n = 33,786), the 10-year CVD risk was estimated using the Framingham Score. The score was calculated as %-risk, with >20% indicating high risk. Overall obesity was measured by BMI, while abdominal obesity was measured by waist circumference. We performed sex-stratified multivariable linear regressions to examine the associations of standardized units of BMI and waist circumference with the 10-year CVD risk, mutually adjusted for waist circumference and BMI.

### Results

Mean (SD) 10-year CVD risks were 14.3(8.9)% in men and 8.0(9.3)% in women, with 37.3% of men and 14.1% of women having high (>20%) risks. After mutual adjustment, one SD in BMI and waist circumference were associated with 0.75(0.50–1.01) and 0.95(0.72–1.18) increase in the %-risk of CVD in men, whereas in women, the β(95% CIs) were 0.43(0.25–0.61) and 1.06(0.87–1.26).

### Conclusion

Abdominal fat accumulation showed stronger associations with 10-year CVD risks than overall adiposity, particularly in women. Although men had higher overall CVD risks, women

Republic of Indonesia. The signed statement prohibits the authors from sharing the data publicly without approval from the Ministry of Health. Data are available from the Ministry of Health, Republic of Indonesia, for researchers who meet the criteria for access to confidential data. The guidelines for data requests can be found at https://layanandata.kemkes.go.id/.

**Funding:** The publication of this paper is funded by the Directorate of Research and Development, Universitas Indonesia, under Hibah PUTI 2024 (Grant Number: NKB-348/UN2.RST/HKP.05.00/2024). The funders had no role in study design, data collection and analysis, decision to publish, or preparation of the manuscript.

**Competing interests:** The authors have declared that no competing interests exist.

experienced more detrimental cardiovascular effects of obesity. Raising awareness of abdominal/visceral obesity and its more damaging cardiovascular effects in women is crucial in preventing CVD-related morbidity and mortality.

## Introduction

Cardiovascular diseases (CVDs) are among the leading causes of death worldwide [1]. In 2019, the World Health Organisation estimated annual cases of 17.9 million deaths due to CVDs, which accounted for 32% of all deaths globally [2]. In Indonesia, CVDs are also among the top three causes of disability-adjusted life years [3]. It is well established that obesity is a risk factor for CVDs, and with more than one-third (35%) of the global adult population having overweight or obesity [4], these individuals are also becoming more susceptible to developing the conditions.

The development of CVDs from obesity involves several simultaneous pathways. A common explanation is elevated cardiac output and eventually blood pressure (hypertension) due to increased demand for oxygen and nutrients from body organs in overall obesity [5]. Chronic hypertension changes the anatomical structure and physiological functions of the heart due to maladaptive remodeling, causing cardiac enlargement and muscle thickening to maintain the ejection fraction, which are causes of CVDs [6]. It is also known that the negative health consequences of obesity are not only influenced by the excess fat mass, but also by the sites in the body where the fat is accumulated [7]. In particular, fat deposition in the abdominal region, known as visceral fat, is associated with detrimental cardiometabolic health outcomes, a pathway that may explain the development of CVDs from abdominal obesity [8]. Visceral fat accumulation in the abdomen leads to liver exposure to high concentrations of free fatty acids and glycerol, which in turn increases hepatic production of triglyceride-rich circulating lipoproteins [9]. This promotes atherosclerosis, or cholesterol plaque buildup in the vascular system, which may result in myocardial infarction or stroke [10]. However, although these possible pathways from overall and abdominal obesity are known, the relative contributions from the two phenotypes of obesity were not often measured quantitatively and comparatively.

To date, large epidemiological studies exploring the associations between body fat distribution and the risk of CVDs in the Asian population were still rare, despite the well-known fact that the Asian populations have higher cardiometabolic risks within the same amount of adiposity compared with the Westerns [11, 12], urging etiological research and public health measures for this population. To what extent overall and abdominal obesity coexist in Asian individuals was also not much known, and whether the separate or joint presence of them is differentially associated with the cardiovascular risks in men and women also remains unclear, despite the differences in body fat distribution between sexes [13, 14], which are generally described by the so-called 'apple-shaped' or 'pear-shaped' adiposity [15]. Therefore, in the present study, we aimed (1) to estimate the 10-year CVD risks in the Indonesian population, and (2) to investigate how the separate and joint presence of overall and abdominal obesity were associated with the 10-year risk.

## Methods

This study is a cross-sectional analysis of the Indonesian National Health Survey, which is a nationwide survey that was conducted by the Ministry of Health, Government of Indonesia. The population-based survey aimed to monitor the general health status of the citizens and

screen for the presence of infectious, cardiometabolic, and degenerative diseases. The survey is held periodically every five years, and in this study which was conducted in July-December 2023, we analysed the 2018 national survey as the most recent one with publicly available data. To ensure the sample representativeness of the general Indonesian population, the national survey applied a multi-stage sampling method combining cluster and systematic random sampling, as well as probability proportional to size methods to select households in the 34 provinces. The national survey also applied weighting factors to account for differences in the populational density and urban-rural distribution across the provinces; hence, the results apply to the whole Indonesian population with its diverse sociodemographic characteristics.

Households were the smallest sampling unit in the national survey, and all individuals or family members were included as participants if their households were selected. Data collection was performed during a household visit, in which healthcare personnel from local Community Health Centres asked participants to complete an interviewer-assisted questionnaire to obtain information on their sociodemographic characteristics, lifestyle, and general health status, including medical history of CVDs. In a weighted and randomly selected subsample of participants, a blood sample was also taken for a glucose and lipid laboratory test. The national survey design and population are described elsewhere in detail [16, 17]. The Health Research Ethics Committee of NIHRD, Ministry of Health, Republic of Indonesia, approved the design of the national survey (LB.02.01/2/KE.267/2017). All participants signed written informed consent, and all data were analysed anonymously. The ethics of this present study have been reviewed and approved by The Research and Community Engagement Ethical Committee, Faculty of Public Health, Universitas Indonesia (No: Ket- 152/UN2.F10.D11/PPM.00.02/2024).

From 1,017,290 individuals who were sampled as participants in the national health survey, in this study we only included non-pregnant individuals aged ≥15 years who were randomly selected for blood laboratory tests (n = 33,786). This random sampling was done because, due to budget and logistic constraints, not all individuals enrolled in the survey underwent the blood glucose and lipid tests [18]. We further excluded those aged <30 years and those with a prior history of CVDs (n = 6,738) [19], as well as those with incomplete biomarkers data (n = 433), resulting in a total number of 26,615 participants. The selection of participants in the present study is illustrated in **S1 Fig**.

## The 10-year CVD risk: The Framingham Risk Score

We calculated the 10-year CVD risk with the Framingham score, as the percent-risk of developing the disease in ten years. The Framingham Risk Score is the most widely used model used by clinicians worldwide to predict the likelihood of CVD incidence in individuals without a prior history of the disease [20]. The score used several clinical characteristics as predictors to estimate the 10-year CVD risk: age, blood pressure, HDL and total cholesterol, and smoking history. It distinguishes the different risks between men and women, giving the two sexes different coefficients for each predictor. The risks of developing CVD in ten years are further stratified into low-risk (<10%), moderate risk (10–20%), and high-risk (>20%). The detailed scoring of the Framingham Risk Score is available in its original publication [19].

## Anthropometric measurements of obesity

To assess the body fat distribution, we used the estimates of overall and abdominal obesity as the study exposures. Overall obesity is measured with BMI, which is an individual's body weight in kilograms divided by the square of his height in meters. The body weight was measured with a digital FESCO ™ weight scale without shoes to the nearest 0.1 kg, whereas the height was measured with a wall-fixed stadiometer to the nearest 0.1 cm. Abdominal obesity is

assessed with waist circumference, which is measured halfway between the iliac crest and the lowest rib with a flexible tape to the nearest 0.1 cm.

Overall obesity was defined as a BMI of $\geq$25.0 kg/m$^2$, according to WHO Asia-Pacific criteria for obesity [21]. Individuals with a BMI ranging from 23.0 to 24.9 kg/m$^2$ were considered overweight, and those with a BMI of <23.0 kg/m$^2$ were considered normal weight. Abdominal obesity was defined as waist circumference above ethnic-specific cut-offs, which are $\geq$90 cm for men and $\geq$80 cm for women in the Asian population [12, 22].

## Clinical biomarkers

From the national survey population, plasma samples from venous blood were taken in a randomly selected and weighted subsample of participants. The puncture was made in the left cubital fossa while the participants were seated. The blood sampling was performed in the morning after the participants were asked to fast overnight. Blood lipid profiles were analysed with the standard chemical chemistry method, while blood glucose levels were measured with a rapid Accu-Check ™ glucometer from a capillary sample. Blood pressure was measured in the left upper arm while the participants were seated after five minutes of rest. The average blood pressure from three measurements was calculated and used in the analyses [16, 17].

## Statistical analyses

Descriptive characteristics were presented as mean (SD) or median (25th–75th percentiles) for continuous variables, and as proportion (%) for categorical variables. As all analyses were weighted to account for the sampling design, the number of participants in the stratified groups could not be obtained, but a percentage was given instead.

To investigate the separate associations of overall or abdominal obesity with the 10-year CVD risks, we performed multivariable linear regressions using the standardised unit of BMI and waist circumference (WC) as the exposures. Effect sizes were presented as regression coefficients (β) with 95% confidence intervals, and the associations were adjusted for sociodemographic confounding factors (age, sex, education, occupation, marital status, and urban/rural living situation).

To investigate which one had a stronger relative contribution to CVD risks, we examined the joint associations of overall obesity (OO) versus abdominal obesity (AO) to the 10-year risks. The association between BMI-CVD risks was mutually adjusted for WC, while the WC-CVD relationship was adjusted for BMI. In addition, we created four combined exposure groups: OO-/AO-(reference), OO+/AO-, OO-/AO+, and OO+/AO+, and multivariable linear regressions were performed to calculate the β (95% CIs) for each group compared to the reference category, adjusted for confounding factors.

To examine whether the associations may differ between sexes, age groups, and urban/rural population settings, we repeated the multivariable linear regression analysis in these subgroups separately. These stratified analyses were done to gain insight into the role of sex and urban/rural differences in the associations between body fat distribution and CVD risks, as previous studies have shown that visceral fat was more strongly associated with insulin resistance and subclinical atherosclerosis in women than men [23, 24], and that urban living situations were associated with higher risks of developing CVD than rural settings [25, 26].

## Results

### Descriptive characteristics of the study population

The sex distribution was similar in the study population (Men: 50.6%; Women: 49.4%), and approximately half of the individuals (54.2%) lived in urban settings. The mean (SD) age of

**Table 1. Descriptive characteristics of the Indonesian population, sex-stratified (n = 26,615).**

| | Total (100%) | Men (50.6%) | Women (49.4%) |
|---|---|---|---|
| Age, *years* | 49.3 (12.3) | 50.2 (11.3) | 48.3 (13.0) |
| Population Setting, *%Urban* | 54.2 | 53.7 | 54.7 |
| Education, *%High* | 5.7 | 6.5 | 5.0 |
| Occupation, *%Unemployed* | 30.2 | 9.2 | 51.7 |
| Marital Status, *%Married* | 85.4 | 90.1 | 80.5 |
| Smoking, *%yes* | 35.2 | 66.2 | 3.3 |
| Systolic Blood Pressure, *mmHg* | 135.8 (24.7) | 134.2 (21.0) | 137.4 (28.4) |
| Diastolic Blood Pressure, *mmHg* | 85.9 (13.2) | 84.5 (11.7) | 87.3 (14.5) |
| HDL Cholesterol, *mg/dL* | 47.6 (11.4) | 44.3 (9.3) | 51.0 (12.5) |
| LDL Cholesterol, *mg/dL* | 125.9 (33.4) | 122.5 (29.1) | 129.3 (37.4) |
| Total Cholesterol, *mg/dL* | 186.4 (39.1) | 180.8 (34.3) | 192.1 (43.0) |
| Triglyceride, *mg/dL* | 137.5 (100.0) | 149.0 (102.5) | 125.6 (90.6) |
| BMI, *kg/m²* | 24.1 (4.7) | 22.9 (3.8) | 25.4 (5.3) |
| Overall Obesity, *%yes* | 38.1 | 26.3 | 50.1 |
| Waist Circumference, *cm* | 81.4 (12.1) | 80.4 (10.8) | 82.4 (13.3) |
| Abdominal Obesity, *%yes* | 39.4 | 20.5 | 58.7 |

Data were presented as mean (SD) for continuous variables, or proportion (%) for categorical variables. All continuous variables described above were normally distributed.

participants was 49.3 (12.3) years. Approximately six-percent participants were highly educated, and they lived more often in urban than rural areas [S1A Table]. Half of women (51.7%) were unemployed or stay-at-home mothers, whereas the unemployment rate was 9.2% for men. Two-thirds of men (66.2%) were smokers, whereas women rarely smoke. Hypertension was common in the population (%), and the mean (SD) of systolic [135.8 (24.7) mmHg] and diastolic [85.9 (13.2)] blood pressure were above normal cut-offs. In regard to the prevalence of obesity, men had only half the prevalence of overall obesity [Men: 26.3%; Women: 50.1%], and one-third of the prevalence of abdominal obesity [Men: 20.5%; Women: 58.7%], than women [Table 1].

The overall and abdominal obesity rates in urban areas were approximately 1.5 times higher than in rural areas. Compared to individuals without obesity, those with (abdominal) obesity have higher average blood pressure levels, total cholesterol, and triglycerides. Other characteristics of the population as stratified by urban/rural living situations and obesity phenotypes were described in **S1A and S1B Table**.

### The estimated 10-year CVD risks

When stratifying the population by sex, men [14.3 (8.9) %] have almost double the risk of developing CVDs in ten years than women [8.0 (9.3) %] of all ages. Among these, 37.3% of men and 14.1% of women have high risks (>20%) of CVDs. When stratifying the population according to urban or rural living situation, the mean (SD) of 10-year CVD risks were not different [Urban: 11.1 (9.4); Rural: 11.3 (10.1) %] in individuals living in urban and rural areas, with similar proportions of those in the high-risk category [Urban: 25.7%; Rural: 25.2%]. In regard to age, more than two-thirds of individuals aged sixty years and older (60–69, 70–70, ≥80) had high risks (>20%) of developing CVDs in ten years. The complete subgroup estimations of the 10-year CVD risks according to sex-, urban/rural-, obesity phenotypes, and age stratifications were presented in **Table 2**.

**Table 2. The stratified 10-year cardiovascular disease risks of the Indonesian population, based on the Framingham Risk Score (n = 26,615).**

| | Total (100%) | Sex | | Living Situation | | Obesity Phenotypes | | | | Age Group (Years) | | | | | |
|---|---|---|---|---|---|---|---|---|---|---|---|---|---|---|---|
| | | Men (50.6%) | Women (49.4%) | Urban | Rural | OO-, AO- | OO+, AO- | OO-, AO+ | OO+, AO+ | 30–39 | 40–49 | 50–59 | 60–69 | 70–79 | ≥80 |
| Mean (SD) 10-Year CVD Risks | 11.2 (9.7) | 14.3 (8.9) | 8.0 (9.3) | 11.1 (9.4) | 11.3 (10.1) | 11.6 (9.5) | 10.5 (8.9) | 11.4 (10.8) | 10.6 (9.9) | 2.9 (2.7) | 7.7 (6.0) | 15.0 (8.8) | 20.4 (8.8) | 24.5 (7.4) | 24.9 (7.4) |
| Proportion (%) with | | | | | | | | | | | | | | | |
| Low Risk | 69.6 | 55.7 | 83.1 | 70.7 | 68.4 | 65.4 | 77.4 | 70.5 | 75.6 | 99.6 | 91.2 | 50.0 | 18.8 | 6.7 | 5.8 |
| Moderate Risk | 4.9 | 7.0 | 2.8 | 3.6 | 6.4 | 8.6 | - | 1.6 | - | 0.1 | 2.1 | 9.1 | 12.3 | 9.8 | 9.7 |
| High Risk | 25.5 | 37.3 | 14.1 | 25.7 | 25.2 | 26.0 | 22.6 | 27.9 | 24.4 | 0.3 | 6.7 | 40.9 | 68.9 | 83.5 | 84.5 |

Data are presented as mean (SD) for continuous variables, or proportion (%) for categorical variables. The 10-year Cardiovascular Diseases risks were calculated using the Framingham Risk Score. The score was calculated as percent-risk (%), in which individuals with <10% were considered low risk, 10–19% as moderate risk, and >20% as having a high risk of developing CVD in ten years.

## The associations of overall and abdominal obesity with the 10-year CVD risks

In both men and women, abdominal adiposity [adjusted β (95% CI): 0.95 (0.72–1.18) in men and 1.06 (0.87–1.24) in women] was more strongly associated with the 10-year CVD risks than overall adiposity [0.75 (0.50–1.01) in men; 0.43 (0.25–0.61) in women] [Table 3]. Additional adjustment for waist circumference strongly attenuated the association between BMI and the 10-year CVD risks, but not so much when the model with waist circumference was mutually adjusted for BMI, implying the higher relative contributions of abdominal obesity to the 10-year CVD risks [Table 3].

After stratifying by urban/rural living situation, this pattern of associations was also observed in both urban and rural populations [S2A Table]. After stratifying by age groups, the strong association between abdominal adiposity and the 10-year CVD risk was also observed in the middle-aged (40–70 years), but not so much in the younger (aged<40) or older (aged>70) adults [S2B Table].

## The joint effect of overall and abdominal obesity on the 10-year CVD Risks

Compared to individuals without any obesity (neither overall nor abdominal obesity) who were set as the reference group, those with both overall and abdominal obesity coexisted had a

**Table 3. The associations of overall and abdominal obesity with the sex-stratified 10-year cardiovascular diseases risks (n = 26,615).**

| | | Total Population | Men | Women |
|---|---|---|---|---|
| **Overall Obesity** | Crude β (95% CI) | -0.70 (-0.84,-0.57) | 0.11 (-0.12,0.34)[#] | 0.17 (0.02–0.31) |
| BMI | Adjusted β (95% CI) | 1.40 (1.29–1.50) | 1.53 (1.35–1.72) | 1.24 (1.12–1.35) |
| SD = 4.7 kg/m$^2$ | +Waist Circumference | 0.67 (0.52–0.82) | 0.75 (0.50–1.01) | 0.43 (0.25–0.61) |
| | | Total Population | Men | Women |
| **Abdominal Obesity** | Crude β (95% CI) | 0.74 (0.60–0.88) | 0.93 (0.72–1.15) | 1.11 (0.95–1.26) |
| Waist Circumference | Adjusted β (95% CI) | 1.41 (1.32–1.50) | 1.46 (1.32–1.60) | 1.39 (1.27–1.51) |
| SD = 12.1 cm | +BMI | 0.92 (0.77–1.07) | 0.95 (0.72–1.18) | 1.06 (0.87–1.24) |

Data were presented as regression coefficient (β) with 95% confidence interval (CI). Multivariate models were adjusted for age, sex, education, occupation, marital status, and living situation (urban/rural). The associations were additionally mutually adjusted for waist circumference and BMI. ([#]) indicates p>0.05. All other associations had p<0.05.

**Table 4. The joint effect of overall and abdominal obesity on the 10-year cardiovascular diseases risks (n = 26,615).**

| | | OO-, AO- (53.2%) | OO+, AO- (7.4%) | OO-, AO+ (8.7%) | OO+, AO+ (30.7) |
|---|---|---|---|---|---|
| **Total Population** | Crude | 1 (Reference) | -1.14 (-1.62, -0.66) | -0.25 (-0.72, 0.22)[#] | -1.04 (-1.34,-0.74) |
| | Adjusted | | 1.44 (1.14–1.74) | 1.82 (1.50–2.15) | 3.00 (2.80–3.21) |
| | | OO-, AO- (70.3%) | OO+, AO- (9.2%) | OO-, AO+ (3.4%) | OO+, AO+ (17.1) |
| **Men** | Crude | 1 (Reference) | -0.89 (-1.53,-0.25) | 4.05 (2.97–5.13) | 2.05 (1.50–2.60) |
| | Adjusted | | 1.46 (1.07–1.85) | 1.73 (0.99–2.47) | 3.20 (2.85–3.55) |
| | | OO-, AO- (35.7%) | OO+, AO- (5.6%) | OO-, AO+ (14.2%) | OO+, AO+ (44.5) |
| **Women** | Crude | 1 (Reference) | -0.80 (-1.38,-0.21) | 2.78 (2.28–3.28) | 1.46 (1.15–1.78) |
| | Adjusted | | 1.48 (1.06–1.91) | 1.96 (1.59–2.33) | 2.78 (2.54–3.02) |

**OO, Overall Obesity; AO, Abdominal Obesity.** Data were presented as regression coefficient (β) with 95% confidence interval (CI). Multivariate models were adjusted for age, sex, education, occupation, marital status, and living situation (urban/rural). ([#]) indicates p>0.05. All other associations had p<0.05.

three-percent higher risk of developing CVDs in ten years [adjusted β (95% CI): 3.00 (2.80–3.21)]. Overall and abdominal obesity coexisted in 30.7% of individuals in the population, whereas the proportions of those who had only overall or abdominal obesity were 7.4% and 8.7%, respectively. Those with only abdominal obesity had a 1.82 (1.50–2.15) higher percent-risk, whereas those with only overall obesity had a 1.44 (1.14–1.74) higher percent-risk, of developing CVDs in ten years compared to those without obesity [**Table 4**].

## Discussion

In this nationally representative study which analyzed large-scale data from the Indonesian National Health survey, we estimated the 10-year CVD risks in the Indonesian population. We also examined the associations of overall and abdominal obesity to the 10-year disease risks, and investigated which obesity phenotype has a stronger association with the cardiovascular risks. We observed that men had almost double the risk than women of developing CVDs in ten years. Men also had more tendencies to be in the high-risk (>20% risks) category than women. On the contrary, overall and abdominal obesity were both more prevalent in women than in men. Both overall and abdominal obesity were associated with the 10-year CVD risks, but the association between overall obesity and the cardiovascular disease risks strongly attenuated after mutual adjustment for abdominal obesity, implying the higher relative contribution of abdominal adiposity in explaining the 10-year CVD risks.

The higher risk of CVD in men, despite the lower prevalence of obesity, may be explained by the fact that men had a smoking habit that was disproportionately higher than women. It is well known from the literature that smoking raises triglycerides and lowers HDL, which is a predisposing factor for atherosclerosis [27, 28]. Previous studies from other populations worldwide also supported the higher burden of CVDs in men than that of women. Men were more affected by premature death due to CVDs, as the risks were higher than women of the same age [29, 30].

Nevertheless, when we look at how obesity impacts CVD risks, this study shows that women suffer more from the detrimental cardiovascular effects of obesity, as they have stronger associations between obesity and CVD risks. This is more particularly so for abdominal than overall obesity, because the association between overall obesity and CVD is strongly attenuated after adjustment for abdominal obesity, which implies that fat accumulation in the abdomen strongly explains the association between overall adiposity and CVD. This sex discrepancy in cardiovascular risks within the same amount of (abdominal) adiposity was also

observed in previous studies, which shows that visceral fat was more strongly associated with insulin resistance and subclinical atherosclerosis in women than men [23, 24].

In the present study, we did not observe a pronounced difference in the 10-year CVD risks between urban versus rural populations, despite the higher prevalence of obesity in urban areas. This may be explained by the fact that, despite urbanization having been strongly linked to increased cardiometabolic risks due to physical inactivity and ultra-processed food consumption, the risks were also rising in the rural population due to older demographics and socioeconomic disadvantages [31, 32]. Obesity was also not differentially related to CVD in urban and rural populations, probably because the sex distribution was similar in the two settings.

The strength of this study is the large sample size from the Indonesian National Health Survey, which enabled us to perform subgroup analyses and supports the generalization of study results to the whole Indonesian population [16, 17]. However, there are several limitations of the present study. First, due to the observational nature of this study, this implied that residual confounding was still possible. However, to lessen this shortcoming, we had adjusted every known and measured confounding factor in the multivariable regression models. Second, although the national survey is held periodically every five years, each survey was cross-sectional by design, as the sample population was randomly selected in each survey. Therefore, no longitudinal time-to-event data was available, and the actual incidence of CVD cannot be examined. Hence, we could only use the estimated 10-year risk from a prediction model. Due to the cross-sectional design, we could not also suggest any causality. However, reverse causation is unlikely as the causal association of obesity to CVD is well established from previous literature [33, 34]. Nevertheless, despite the mentioned limitations, this study is still the most representative of the general Indonesian population, as extensive cohort studies in Indonesia are still rare, particularly with a population as large as this survey.

This study has several public health and research implications. First, the stronger relative contribution of abdominal obesity over overall obesity to the CVD risks implies the importance of highlighting abdominal obesity in preventing CVDs. This poses a challenge as an earlier study has shown that the general public is less aware of the meaning of abdominal obesity and the negative health consequences that pertain to it [35]. Hence, raising public awareness on the definition, measurement, and prevention of abdominal obesity to reduce the risk of CVD in the population is crucial, and routine measurement of waist circumference in daily clinical practice may become a practical but fundamental starting point. Nevertheless, a previous study revealed that waist circumference measurement was not performed routinely by general practitioners in their daily clinical practices, and thus, reinforcing its routine measurements is necessary [36]. Second, as this study revealed the more detrimental cardiovascular effect of (abdominal) obesity in women than in men, attention or targeted intervention to prevent obesity and its impact on women should be prioritized, especially because earlier studies have shown the higher prevalence of obesity in adult women than men, particularly among the Asian population [37, 38]. Sex differences should also be taken into account when designing public health policies related to obesity and CVDs. Lastly, as in this study, we could only measure abdominal obesity as a proxy for visceral adiposity; we also recommend further (longitudinal) studies with direct measure of visceral fat (e.g., with MRI or CT scan) to straightly investigate how visceral adiposity impacts CVD risks differentially in men and women.

Taken together, to conclude this nationally representative study, we observed that although men had almost double the risk than women of developing CVD in ten years, women had a higher prevalence of obesity and suffered more from the detrimental cardiovascular effects of obesity than men. Abdominal obesity, presumably explained by visceral fat accumulation in the abdomen, was more strongly associated with an increased risk of CVD than overall

adiposity, and more particularly so in women than men. Raising awareness of the importance of abdominal obesity in clinical practice, along with public health policies to address the more damaging cardiovascular effects of abdominal/visceral adiposity in women than in men, may be beneficial in preventing CVD and its related morbidity and mortality. As this study is particularly relevant to the Indonesian population, similar studies in other countries may be necessary to generalize the results to the broader global population.

## Supporting information

**S1 Fig. Study flow chart.**
(DOCX)

**S1 Table. a and b.** Descriptive Characteristics of the Indonesian Population, as Stratified by Urban/Rural Living Situation and Obesity Phenotypes (n = 26,615).
(DOCX)

**S2 Table. a and b.** The Associations of Overall and Abdominal Obesity with the 10-Year Cardiovascular Diseases Risks, as Stratified by Urban/Rural Living Situation and Age Group (n = 26,615).
(DOCX)

## Author Contributions

**Conceptualization:** Fathimah S. Sigit, Dicky L. Tahapary, Woro Riyadina.

**Data curation:** Fathimah S. Sigit, Dicky L. Tahapary, Woro Riyadina.

**Formal analysis:** Fathimah S. Sigit, Dicky L. Tahapary, Kusharisupeni Djokosujono.

**Funding acquisition:** Fathimah S. Sigit.

**Investigation:** Fathimah S. Sigit, Dicky L. Tahapary, Woro Riyadina, Kusharisupeni Djokosujono.

**Methodology:** Fathimah S. Sigit, Kusharisupeni Djokosujono.

**Project administration:** Fathimah S. Sigit.

**Resources:** Fathimah S. Sigit, Woro Riyadina.

**Software:** Fathimah S. Sigit.

**Supervision:** Dicky L. Tahapary, Woro Riyadina, Kusharisupeni Djokosujono.

**Visualization:** Fathimah S. Sigit.

**Writing – original draft:** Fathimah S. Sigit.

**Writing – review & editing:** Dicky L. Tahapary, Woro Riyadina, Kusharisupeni Djokosujono.

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
