## [Decision Letter · Decision Letter 0]

18 Jun 2024

PONE-D-24-16324Sex Disparities in the Associations of Overall versus Abdominal Obesity with the 10-Year Cardiovascular Disease Risk: Evidence from the Indonesian National Health SurveyPLOS ONE

Dear Dr. Sigit,

Thank you for submitting your manuscript to PLOS ONE. After careful consideration, we feel that it has merit but does not fully meet PLOS ONE’s publication criteria as it currently stands. Therefore, we invite you to submit a revised version of the manuscript that addresses the points raised during the review process.

We look forward to receiving your revised manuscript.

Kind regards,

Emmanuel Kwaku Ofori, PhD

Academic Editor

PLOS ONE

Journal Requirements:

"The publication of this paper is funded by the Directorate of Research and Development, Universitas Indonesia, under Hibah PUTI 2024."

4. In the online submission form, you indicated that "The dataset analysed in this study is available on reasonable request from the Ministry of Health, Republic of Indonesia, subject to prior review."

Reviewers' comments:

Reviewer's Responses to Questions

**Comments to the Author**

1. Is the manuscript technically sound, and do the data support the conclusions?

Reviewer #1: Partly

Reviewer #2: Yes

2. Has the statistical analysis been performed appropriately and rigorously? 

Reviewer #1: Yes

Reviewer #2: Yes

3. Have the authors made all data underlying the findings in their manuscript fully available?

Reviewer #1: No

Reviewer #2: Yes

4. Is the manuscript presented in an intelligible fashion and written in standard English?

Reviewer #1: Yes

Reviewer #2: Yes

5. Review Comments to the Author

Reviewer #1: Although the statistical analyses performed for this work are appropriate, it is difficult to affirm if the results support the conclusion because the authors did not report the respective p-values associated with the regression analyses. The authors need to report the p-values in order to make all the relevant data available without restriction.

Reviewer #2: 1. The manuscript is technically sound using appropriate statistical analyses, including the Framingham Risk Score and sex-

stratified multivariable linear regressions on nationally representative data from the Indonesian Health Survey. The

conclusions are well supported by the data presented.

2. Statistical analysis have been performed appropriately and rigorously.

3. All data underlying the findings in their manuscript are available.

4. The manuscript is presented clearly and written in standard English. It follows a logical structure ensuring coherent

presentation of the research.

6. PLOS authors have the option to publish the peer review history of their article (what does this mean?). If published, this will include your full peer review and any attached files.

Reviewer #1: No

Reviewer #2: **Yes: **Nathaniel Ebo Aidoo

---

## [Author Response · Author response to Decision Letter 0]

3 Jul 2024

We thank the editor and reviewers for the opportunity to revise and resubmit our manuscript. We found the feedback from the editor and reviewers constructive and helped us develop our manuscript further. Please find below our point-by-point responses addressing each comment raised by the reviewers.

Reviewer #1: 

1. Although the statistical analyses performed for this work are appropriate, it is difficult to affirm if the results support the conclusion because the authors did not report the respective p-values associated with the regression analyses. The authors need to report the p-values in order to make all the relevant data available without restriction. 

------

We thank Reviewer #1 for his/her thorough feedback and constructive criticism of our manuscript. However, following the STROBE guideline to report observational/epidemiological studies, we have properly presented the results of our study in regression coefficients and 95% CIs to show the magnitude of effect (effect size) and its measure of precision. The 95% CI is a robust indicator of the degree of error, as those that do not cross 0 mean they are statistically consistent (always positive if β is positive, always negative when β is negative). Using the p-value is not recommended due to its dichotomizing properties, which could potentially under- or over-estimate the results, and its lack of clinical significance (see references below). Nevertheless, to convince the reviewer of our results, we have now also included the significance of all effect sizes (p < or > 0.05). Solidifying our conclusion, we found that the majority of p-values from the linear regression analysis were <0.05. Only two crude associations had p>0.05 (in which the 95% CIs also crossed 0), and they became significant (p<0.05) when the associations were adjusted for confounding factors.

Reference: 

• von Elm E, Altman DG, Egger M, et al. The Strengthening the Reporting of Observational Studies in Epidemiology (STROBE) statement: guidelines for reporting observational studies. Lancet. 2007;370(9596):1453-1457. 

• Sullivan GM, Feinn R. Using Effect Size-or Why the P Value Is Not Enough. J Grad Med Educ. 2012;4(3):279-282. 

• Greenland S, Senn SJ, Rothman KJ, et al. Statistical tests, P values, confidence intervals, and power: a guide to misinterpretations. Eur J Epidemiol. 2016;31(4):337-350.

• Ioannidis, J. P. A. (2019). What Have We (Not) Learnt from Millions of Scientific Papers with P Values? The American Statistician, 73(sup1), 20–25. 

• Wasserstein, R. L., & Lazar, N. A. (2016). The ASA Statement on p-Values: Context, Process, and Purpose. The American Statistician, 70(2), 129–133. 

2. “Using the large, nationally representative data from the Indonesian National Health Survey, we analysed the population as stratified by sex, age, and urban/rural characteristics to gain insight into how these characteristics play a part in the associations between overall or abdominal obesity and CVD risks.”  This information is better suited for the methods section.

------

Changed accordingly. We have moved this part to the Methods section and incorporated it with the ‘Statistical Analyses’. Line 183, Page 7, highlighted in yellow in the manuscript:

“To examine whether the associations may differ between sexes, age groups, and urban/rural population settings, we repeated the multivariable linear regression analysis in these subgroups separately. These stratified analyses were done to gain insight into the role of these characteristics in the associations between body fat distribution and CVD risks.”

3. “The survey is held periodically every five years, with the 2018 survey being the most recent one with publicly available data, which we analysed in this study.”  Within which period was this study carried out?

------

We have now elaborated the information in the manuscript. Line 100, Page 4, highlighted in yellow:

“The survey is held periodically every five years, and in this study which was conducted in July-December 2023, we analysed the 2018 national survey as the most recent one with publicly available data.”

4. “From 1,017,290 individuals who were sampled as participants in the national health survey, in this study we only included non-pregnant individuals aged >15 years who were randomly selected for blood laboratory tests (n=33,786).”  How did the authors arrive at this figure? What were the assumptions for sample size calculation?

------

We used the data of all participants included in the national health survey, who were already weighted and randomly sampled for blood laboratory tests, to represent the whole Indonesian population. This random sampling was done because, due to budget and logistic constraints, not all individuals enrolled in the survey underwent the blood glucose and lipid tests.

Hence, the sampling design and calculation in this study follow the original national survey, which employed multi-stage systematic random sampling and probability proportional to size methods, to take into account differences in geographical density and urban-rural distribution in the 34 provinces, as we have mentioned in the manuscript. The detailed sampling steps of the Indonesian National Health Survey, including the census block selection at the regional and national level, were described in detail in the government’s official report, which we also now referenced in our manuscript. We also further clarified the sample inclusion in the study flow chart. 

Reference:

Research & Development Organization, Ministry of Health, Republic of Indonesia (2019). Report on the national basic health research. Accessed 24 June 2024, <https://repository.badankebijakan.kemkes.go.id/id/eprint/3514/1/Laporan%20Riskesdas%202018%20Nasional.pdf>.

Line 124, Page 5, highlighted in yellow in the manuscript

“From 1,017,290 individuals who were sampled as participants in the national health survey, in this study we only included non-pregnant individuals aged >15 years who were randomly selected for blood laboratory tests (n=33,786). This random sampling was done because, due to budget and logistic constraints, not all individuals enrolled in the survey underwent the blood glucose and lipid tests.”

5. “We further excluded those aged <30 years and those with a prior history of CVDs (n=6,738), ...”  Arbitrarily setting a threshold of age 30 raises the potential of bias unless the authors can provide reference(s) to support this.

------

We used the cut-off of 30 years to follow the estimation criteria of the Framingham Risk Score (FRS). According to the original publication of FRS, the risk score was meant to be used in adults >30 years, as the Framingham Heart Study only included those aged >30. Furthermore, the estimated 10-year risk of cardiovascular events can be calculated in age categories of 30-34, 35-39, 40-44, 45-49, 50-54, 55-59, 60-64, 65-69, 70-74, and >75 years, with different coefficients/points given for each age category in the prediction model. We also followed this categorization in the age-stratified analysis of our study. We have now properly cited the original publication in the manuscript.

Reference:

-D'Agostino RB Sr, Vasan RS, Pencina MJ, et al. General cardiovascular risk profile for use in primary care: the Framingham Heart Study. Circulation. 2008;117(6):743-753. doi:10.1161/CIRCULATIONAHA.107.699579 

-Dyslipidemia Guidelines Tool, Canadian Cardiovascular Society, 2017

6. “From the national survey population, blood venous plasma samples were taken in a randomly selected and weighted subsample of participants.”  plasma from venous blood?

------

We thank the reviewer for noticing this. We have now rephrased this sentence for better clarity. 

Line 156, Page 6, highlighted in yellow in the manuscript

“From the national survey population, plasma samples from venous blood were taken in a randomly selected and weighted subsample of participants.” 

7. “Abdominal obesity, presumably explained by visceral fat accumulation in the abdomen, was more strongly associated with an increased risk of CVD than overall adiposity, and more particularly so in women than men. Raising awareness of the importance of abdominal obesity in clinical practice may benefit in preventing CVD and its related morbidity and mortality. Further studies and public health policies to address the more damaging cardiovascular effects of abdominal/visceral adiposity in women than in men are necessary.”  The authors must admit limitations of generalizability as the study strictly focuses on the Indonesian population.

------

We have now added the suggestion from the reviewers accordingly. 

Line 323, Page 12, highlighted in yellow in the manuscript

“As this study is particularly relevant to the Indonesian population, similar studies in other countries may be necessary to generalize the results to the broader global population.” 

8. “Data were presented as mean (SD) for continuous variables, or proportion (%) for categorical variables.”  The authors indicated under the statistics section of the methods that descriptive statistics were presented including median (25-75th percentiles). Authors must indicate which variables were reported as such.

------

We thank the reviewer for this comment. We described all continuous variables as mean (SD) because, after inspecting the histogram, all variables were normally distributed (have a bell-shaped curve). We have added this information to the legend of Table 1. 

Legend of Table 1, Line 461, Page 16, highlighted in yellow in the manuscript

“Data were presented as mean (SD) for continuous variables, or proportion (%) for categorical variables. All continuous variables described above were normally distributed.“

Reviewer #2: 

1. The manuscript is technically sound using appropriate statistical analyses, including the Framingham Risk Score and sex-stratified multivariable linear regressions on nationally representative data from the Indonesian Health Survey. The conclusions are well supported by the data presented.

2. Statistical analysis have been performed appropriately and rigorously. 

3. All data underlying the findings in their manuscript are available. 

4. The manuscript is presented clearly and written in standard English. It follows a logical structure ensuring coherent presentation of the research.

------

We are deeply grateful to Reviewer #2 for his/her overall positive response to our manuscript.

---

## [Editor Report · Decision Letter 1]

16 Jul 2024

Sex Disparities in the Associations of Overall versus Abdominal Obesity with the 10-Year Cardiovascular Disease Risk: Evidence from the Indonesian National Health Survey

PONE-D-24-16324R1

Dear Dr. Sigit,

We’re pleased to inform you that your manuscript has been judged scientifically suitable for publication and will be formally accepted for publication once it meets all outstanding technical requirements.

Kind regards,

Emmanuel Kwaku Ofori, PhD

Academic Editor

PLOS ONE

Additional Editor Comments (optional):

Manuscript is improved. All queries and suggestions have been effected by the authors.
---

## [Editor Report · Acceptance letter]

5 Aug 2024

PONE-D-24-16324R1 

PLOS ONE

Dear Dr. Sigit, 

I'm pleased to inform you that your manuscript has been deemed suitable for publication in PLOS ONE. Congratulations! Your manuscript is now being handed over to our production team.

Kind regards, 

on behalf of

Dr Emmanuel Kwaku Ofori 

Academic Editor

PLOS ONE